# Low-Dose Neubotulinum Toxin A versus Low-Dose Abobotulinum Toxin A Injection for the Treatment of Cervical Dystonia: A Multicenter, 48-Week, Prospective, Double-Blinded, Randomized Crossover Design Study

**DOI:** 10.3390/toxins13100694

**Published:** 2021-10-01

**Authors:** Subsai Kongsaengdao, Arkhom Arayawithchanont, Kanoksri Samintharapanya, Pichai Rojanapitayakorn, Benchalak Maneeton, Narong Maneeton

**Affiliations:** 1Division of Neurology, Rajavithi Hospital, Department of Medical Services, Ministry of Public Health, Bangkok 10400, Thailand; 2Division of Neurology, Department of Medicine, College of Medicine, Rangsit University, Bangkok 10400, Thailand; 3Division of Neurology, Department of Medicine, Sunpasitthiprasong Hospital, Ubon Ratchathani 34000, Thailand; aarayawi@gmail.com; 4Division of Neurology, Department of Medicine Lampang Hospital, Lampang 52000, Thailand; tem2000_1@yahoo.com; 5Division of Neurology, Department of Medicine Suratthani Hospital, Suratthani 84000, Thailand; pichairoj2518@gmail.com; 6Department of Psychiatry, Faculty of Medicine, Chiang Mai University, Chiang Mai 50200, Thailand; benchalak.maneeton@cmu.ac.th (B.M.); narong.m@cmu.ac.th (N.M.)

**Keywords:** abobotulinum toxin A, neubotulinum toxin A, the Toronto Western Spasmodic Torticollis Rating Scale, the Cervical Dystonia Impact Profile, the Short Form 36 health survey questionnaire, the Center for Epidemiological Studies-Depression Scale, the Patient Health Questionnaire-9

## Abstract

Various types of botulinum toxin (BoNT) have been studied to treat cervical dystonia (CD). Although high-dose BoNT has proven efficacy, it increases the risk of adverse events. For this reason, this study was planned to identify the non-inferiority efficacy, tolerability, and safety of low-dose neubotulinum toxin A (Neu-BoNT-A) versus low-dose abobotulinum toxin A (Abo-BoNT-A) in CD treatment. The 48-week, prospective, randomized, controlled crossover design study of CD treatment, with 50-unit Neu-BoNT-A and 250-unit Abo-BoNT-A injections at 12-week intervals, was conducted over a 24-week treatment period. This study used the following standardized rating scales to assess the efficacy of BoNT: the Toronto Western Spasmodic Torticollis Rating Scale (TWSTRS); health-related quality of life (HRQoL); the Cervical Dystonia Impact Profile (CDIP-58); the Short Form 36 health survey questionnaire (SF-36); and, for the depressive symptoms of CD patients, the Center for Epidemiological Studies-Depression Scale (CES-D) and the Patient Health Questionnaire-9 (PHQ-9). Fifty-two CD patients were enrolled from October 2019 to January 2021. The mean scores of the TWSTRS total at the post-treatments in both Neu-BoNT-A and Abo-BoNT-A had a significant reduction from baseline (*p* = 0.008 and 0.002, respectively). However, the mean changes of the TWSTRS total at the 12- and 24-week treatments between the two treatment groups were not significantly different (*p* = 0.284 and 0.129, respectively). The mean scores of the HRQoL questionnaires (the CIDP-58 and the SF-36) and the depressive symptoms (the CES-D and the PHQ-9) in both treated groups at the post-treatments did not significantly decrease from baseline and were comparable. Two patients treated with Abo-BoNT-A (250 units) reported cervical tension and benign paroxysmal positional vertigo (BPPV). There were no serious adverse events reported. Though both low-dose BoNT-As were effective at improving clinical symptoms without significant side effects, both treatments did not predict change in quality of life and depression. With the non-inferiority criteria, low-dose Neu-BoNT-A has a similar efficacy, safety, and tolerability to Abo-BoNT-A.

## 1. Introduction

Cervical dystonia (CD), a common abnormal movement disorder in which patients suffer from repetitive and/or sustained involuntary contractions of the neck muscles, resulting in abnormal neck twisting and/or posture, could negatively affect quality of life and occupational and social functions. Therefore, the effective treatment of CD, including botulinum toxin (BoNT) injections, can improve said quality of life and functions. The medical and cosmetic fields widely use BoNT. Since BoNT blocks acetylcholine releasing at the neuromuscular junctions, it could inhibit muscle contraction and decrease spastic muscle tone [1]. Five botulinum toxins [2] approved for the treatment of cervical dystonia include onabotulinum toxin A (Ona-BoNT-A, BOTOX), abobotulinum toxin A (Abo-BoNT-A, Dysport), incobotulinum toxin A (Inco-BoNT-A, Xeomin), neubotulinum toxin A (Neu-BoNT-A, Neuronox), and rimabotulinum toxin B (Rima-BoNT-B, Myobloc).

According to a double-blind, randomized, controlled trial, both BoNT-A (25th to 75th percentile range of 198–300 units) and BoNT-B showed efficacy in reducing the CD-specific impairments measured by the Toronto Western Spasmodic Torticollis Rating Scale (TWSTRS), mainly in severity [2]. The botulinum toxin type A and B [3,4] injections were equally effective and safe in adult CD patients, whereas side effects such as dry mouth and sore throat from BoNT-B were higher than from BoNT-A treatment [5]. Of those BoNTs, Ona-BoNT-A is the best known, dominating the botulinum toxin market since it was first approved and marketed in the United States in 1989, and Abo-BoNT-A has ranked second in the toxin market [6]. An alternative BoNT, Neu-BoNT-A, structurally similar to Ona-BoNT-A, could reduce the cost of treatment and be convenient in dosing and safety [6]. The only difference between the two brands is that Ona-BoNT-A is a vacuum-dried preparation, whereas Neu-BoNT-A is freeze-dried [7]. Neubotulinum toxin A (Neuronox^®^) is a 900 KDa complex of neurotoxins from the Hall A Hyper strain of *Clostridium botulinum* and freeze dry-lyophilized powder with albumin stabilizer added (toxin–hemagglutinin complex), whereas abobotulinum toxin A (Dysport^®^) is a 500–900 KDa complex of neurotoxins from the ATCC 3502 Hall A strain of *Clostridium botulinum* and freeze dry-lyophilized powder with albumin stabilizer added (toxin–hemagglutinin complex), and onabotulinum toxin A (Botox^®^) is a 900 KDa complex of neurotoxins from the Hall A Allergan strain of *Clostridium botulinum* and freeze dry-vacuum powder with albumin stabilizer added (toxin–hemagglutinin complex).

A previous multicenter, randomized, controlled trial illustrated that Ona-BoNT-A and Neu-BoNT-A have comparable efficacies and safety by using 4 U/kg for hemiplegia and 6 U/kg for diplegia in treating spastic equinus in children with cerebral palsy [8] and benign essential blepharospasm [7]. Even though BoNT is beneficial in CD treatment, some patients finally stop BoNT treatment permanently. Reasons for discontinuation include unsuccessful treatment, poor adherence, difficulty accessing the treatment (e.g., social, transport, and financial problems) [9], and side effects.

Accordingly, we conducted a double-blind, randomized crossover design study to evaluate the efficacy, tolerability, quality of life, and depressive symptoms of low doses of two types of botulinum toxins (Neu-BoNT-A and Abo-BoNT-A; conversion ratio of 1:5/Neu-BoNT-A (100 units): Abo-BoNT-A (500 units)) in the treatment of cervical dystonia.

## 2. Results

A total of 52 CD patients (female = 36 and male = 16) were enrolled from October 2019 to January 2021. The mean (SD) age of the patients was 52.4 (14.2) years, and the mean (SD) age of first cervical dystonia diagnosis was 49.0 (13.6) years. The mean (SD) duration of cervical dystonia was 3.1 (3.1) years, with a range of 0–14 years, and the mean (SD) duration of previous cervical dystonia treatment was 2.8 (2.7) years. Forty-five patients were previously treated with Abo-BoNT-A, one patient was previously treated with Neu-BoNT-A, and six patients were never treated with BoNT-A. All the enrolled patients were included in the analysis. The clinical presentation of the patients consisted of torticollis (*n* = 35), laterocollis (*n* = 2), anterocollis (*n* = 4), retrocollis (*n* = 4), and mixed type (*n* = 9). No patients had any other significant medical conditions.

### 2.1. Primary Outcomes

#### 2.1.1. The Clinical Outcome of CD

The mean scores of the TWSTRS Total of the Neu-BoNT-A- and Abo-BoNT-A-treated groups were significantly reduced from baseline (*p* = 0.008 and 0.002, respectively) (see Table 1). However, the mean changes from baseline were not significantly different between the two treatment groups after the 12- and 24-week treatments (*p* = 0.284 and 0.129, respectively) (see Table 2). Considering each subscale, the mean scores of the TWSTRS severity subscale in both groups were significantly decreased from baseline (*p* = 0.020 and 0.009, respectively) (see Table 1 and Figure 1), and the mean change from baseline in the Abo-BoNT-A-treated group was reduced with a significant difference from Neu-BoNT-A after the 24-week treatments (*p* = 0.034) (see Table 2 and Figure 2). The mean of the TWSTRS disability subscale significantly decreased from baseline in only the Neu-BoNT-A-treated group (*p* = 0.014) (see Table 1 and Figure 1). The mean of the TWSTRS pain subscale was significantly decreased in only the Abo-BoNT-A-treated group (*p*
*=* 0.015) (see Table 1 and Figure 2).

#### 2.1.2. The Disease-Specific Health-Related Quality of Life

The mean scores of CIDP-58 were not significantly decreased from baseline in either group (*p* = 0.232 and 0.416, respectively) (see Table 1 and Figure 1). Similarly, the mean changes from baseline between the two groups were not significantly different after the 12- and 24-week treatments (*p* = 0.740 and 0.240, respectively) (see Table 2 and Figure 1).

### 2.2. Secondary Outcomes

#### 2.2.1. The General Health-Related Quality of Life

The mean scores of the SF-36 were not significantly decreased from baseline in either group (*p* = 0.731 and 0.911, respectively) (see Table 1 and Figure 1). Similarly, the mean changes of the SF-36 scores from baseline between the two groups were not significantly different after the 12- and 24-week treatments (*p* = 0.460 and 0.440, respectively) (see Table 2 and Figure 2)**.**

#### 2.2.2. Depressive Symptoms

The study showed that the mean scores of the CES-D were not significantly decreased from baseline in the two groups (*p* = 0.573 and 0.726, respectively) (see Table 1 and Figure 2). Accordingly, the mean changes of the CES-D scores from baseline were not significantly different between the two groups after the 12- and 24-week treatments (*p* = 0.072 and 0.922, respectively) (see Table 2 and Figure 1). In addition, the mean scores of the PHQ-9 in both treatment groups were not significantly decreased from baseline (*p* = 0.226 and 0.633, respectively) (see Table 1 and Figure 1). Moreover, the mean changes of the PHQ-9 scores from baseline were not significantly different between the two groups after the 12- and 24-week treatments (*p* = 0.760 and 0.320, respectively) (see Table 2 and Figure 1).

### 2.3. Adverse Events

Two patients treated with Abo-BoNT-A (250 units) reported cervical tension and benign paroxysmal positional vertigo (BPPV). The patients treated with Neu-BoNT-A (50 units) had no adverse events. There were no serious adverse events reported.

## 3. Discussion

According to the findings of this study, both Neu-BoNT-A (50 units) and Abo-BoNT-A (250 units) were effective in reducing overall CD symptoms in the 24-week treatment. According to a non-inferiority statistical analysis comparing the mean changes from baseline, low doses of the two BoNT-As were not different (beta error = 0.9 and alpha error = 0.05) in the reduction in overall CD symptoms in 12- and 24-week treatments. Although both low-dose BoNT-As could decrease the severity symptoms, only a Neu-BoNT-A could decrease disability, and Abo-BoNT-A could decrease pain after the 24-week treatment. Neither BoNT-A treatment improved the disease-specific or general HRQoL, and neither increased or decreased depressive symptoms after the post-treatment period. Interestingly, a low dose of either BoNT-A had rare adverse events in the treatment of CD.

The present study found that low doses of Abo-BoNT-A were effective in treating CD symptoms according to the TWSTRS, especially the severity subscore, as previously evidenced [10,11,12]. Since the efficacies of Abo-BoNT-A and Neu-BoNT-A in the present study were comparable, an alternative treatment with Neu-BoNT-A in CD patients could be conducted in clinical practice.

There have been few studies evaluating the disease-specific HRoQL of CD patients after BoNT-A treatment [13,14]. An open-label study of twenty CD patients after a 6-week treatment with Neu-BoNT-A injections illustrated an improvement of HRoQL, measured by the craniocervical dystonia questionnaire (CDQ-24) [10]. Similarly, an open-label study of 24 weeks with three injections of 50-unit Neu-BoNT-A suggested an improvement of disease-specific HRoQL, measured by the CIDP-58 and the CDQ-24, in CD patients [13]. Additionally, our previous open-label study also displayed an improvement of disease-specific HRoQL, measured by the CDIP-58, in 20 CD patients treated with eight injections of 250-unit Abo-BoNT-A in 3-month intervals over a 2-year treatment period [15]. A previous study of a 4-week cycle of treatment with high-dose Abo-BoNT-A (500 units) displayed an improvement of disease-specific HRoQL in CD patients [16]. Compared with previous results, this low-dose BoNT-A treatment did not show a significant change in HRoQL. Therefore, a further well-defined and large sample size study should be conducted to verify those findings.

There are limited studies of the various efficacies of BoNT-A for the improvement of general HRoQL in CD patients. The previous short- and long-term, open-label studies did not find any benefits of BoNT-A for the improvement of general HRoQL in CD patients [13,17]. However, an RCT of a 500-unit Abo-BoNT-A treatment of CD has illustrated an improvement of general HRoQL measured by the SF-36 [12]. Another RCT using either Abo-BoNT-A (mean dose = 557 units) or Botox (mean dose = 115.7 units) also showed an improvement of general HRoQL, measured by the SF-36 [11]. Hence, the absence of an improvement of general HRoQL in our present RCT may be caused by the use of a lower-dose treatment of BoNT-A compared to the previous RCTs.

A previous meta-analysis suggested that BoNT-A can treat depressive symptoms [18]. However, another study illustrated that BoNT-A treatment did not improve depression, measured by the Beck Depression Scale (BDS) [19]. The result of our study was that treating with low-dose BoNT-A did not affect the score of depression, as in the latter study.

The treatment of CD patients with BoNT-A is associated with an increased risk of adverse events, particularly dysphagia, dry mouth, and weakness [20,21]. Unfortunately, no previous RCTs have determined the safety of multiple cycles of injection of BoNT-A [21]. To our knowledge, this is the first study that has reported on the safety of low-dose Neu-BoNT-A and Abo-BoNT-A injections for CD treatment, which have a low rate of adverse events, especially in 50-unit Neu-BoNT-A injections.

## 4. Conclusions

In summary, the low doses of Neu-BoNT-A and Abo-BoNT-A improved CD symptoms after the 12- and 24-week treatments. Unfortunately, their effects could not decrease the depressive symptoms and could not improve the disease-specific and general HRoQL in the CD patients. Based on the non-inferiority criteria, low doses of Neu-BoNT-A and Abo-BoNT-A (conversion ratio of 1:5/Neu-BoNT-A (100 units): Abo-BoNT-A (500 units)) are comparable in terms of efficacy, tolerability, and quality of life for the treatment of CD patients.

## 5. Materials and Methods

This study was a phase III, multicenter, 48-week, prospective, double-blinded (the participants, care providers, investigators, and outcomes assessors), randomized crossover design study. We aimed to compare the efficacy and the safety of low-dose Neu-BoNT-A and Abo-BoNT-A to treat cervical dystonia (CD). This study was carried out between October 2019 and January 2021. The research protocol was approved by the Ethics Committee of the Ministry of Public Health of Thailand, and the researchers performed the study according to the Declaration of Helsinki and the International Conference on Harmonization/Good Clinical Practice Guidelines. Written informed consent was obtained from all participants. The study was registered at ClinicalTrail.gov (identifier: NCT03805152).

Inclusion criteria: We invited all eligible patients aged 18 or more, fulfilling the diagnostic criteria for primary cervical dystonia, providing informed consent, having normal consciousness, having good communication, and understanding the Thai language to participate in the study. The investigator would instruct all healthy and sexually active female subjects to avoid pregnancy during the study period. These participants also had negative urine pregnancy tests before including themselves in the study. All eligible patients had to cooperate with physical and neurological examinations during the whole study period.

Exclusion criteria: the exclusion criteria were:Unable or unwilling to comply fully with the protocol.Pregnant, lactating, or at risk of pregnancy during the study and not taking adequate precautions against pregnancy.Conditions which could influence the clinical trial:
Medical conditions: bleeding abnormalities, thrombocytopenia, arthritis, heart disease, and history of botulism.Neurological conditions: other neuromuscular disorders (e.g., myasthenia gravis, Lambert–Elton Syndrome), and dementia.Psychiatric conditions except for depressive disorder (e.g., psychotic spectrum, dementia).Known history of drug abuse (narcotic(s), Cafergot (ergotamine/caffeine), or others) or drug allergy.Known hypersensitivity to any of the test materials, related compounds, or BoNT-A.Received any unlicensed drug within the previous six months or treated with the investigational drug(s) within six months before the screening visit.Previously entered in this study.Unable to cooperate with the follow-up neuropsychological test.Planned to schedule elective surgery during the study.Using aminoglycoside antibiotics or curare.

At the screening visit, the investigators would explain the details of the study. We assessed the patient’s medical history and their recent and current medication. We performed a complete physical examination and assessed the hematological test (complete blood count) and the urine pregnancy test before the injection of botulinum toxin A.

### 5.1. Study Medication

This crossover intervention compared low doses of two types of BoNT-A: Neu-BoNT-A (Neuronox^®^) and Abo-BoNT-A (Dysport^®^) with a conversion ratio of 1:5 (Neu-BoNT-A 100 units: Abo-BoNT-A 500 units). Both BoNT-As were freeze-dried powders, diluted in normal saline, and used within two hours of preparation. Neu-BoNT-A (100 units) and Abo-BoNT-A (500 units) were diluted in 6.0 mL of normal saline solution, resulting in approximately 16.67 units of Neu-BoNT-A per 1.0 mL and 83.33 units of Abo-BoNT-A per 1.0 mL. All enrolled patients were randomly assigned to receive a low dose of either Neu-BoNT-A (50 units) or Abo-BoNT-A (250 units). Neurologists assessed the dystonic position and the agonist and antagonist muscle spasms of the sternocleidomastoid, trapezius, and splenius capitis muscles to determine the injection site. We screened and assessed the patients and provided the intramuscular injections of the neck muscles at the first visit and at every 12-week interval over a 24-week treatment period, as shown in Figure 2. After the first 24-week treatment period, we switched the patients to another treatment arm for another 12-week interval over a 24-week treatment period.

### 5.2. Assessments

During the evaluation of the participants, we blinded all patients, assessors, and investigators. The TWSTRS and questionnaires, including the CDIP-58, SF-36, PHQ-9, and CES-D, were applied. All questionnaires were thoroughly explained to all patients. The primary outcome was the change in mean TWSTRS and CDIP-58 scores from pre-treatment to 24 weeks after two injections of Abo-BoNT-A or Neu-BoNT-A. The secondary outcomes were the mean changes of SF-36, PHQ-9, and CES-D scores (see in Figure 1).

#### 5.2.1. Primary Outcome Measures

##### Efficacy Measurement

We measured the clinical outcome [22,23,24]. The TWSTRS, a validated scale, has been frequently applied in clinical trials [25]. The TWSTRS [26] ranged from 0 to 85, by summation of all three subscales. The higher the score, the more severe the disease. The TWSTRS subscales include torticollis severity scale or TWSTRS-Severity Scale (range of 0–35 points), TWSTRS-Disability Scale (range of 0–30 points), and TWSTRS-Pain Scale (range of 0–20 points). The inter-rater agreement follows the guidelines of the TWSTRS videotape protocol [27].

##### Disease-Specific HRQoL

The disease-specific HRQoL [28] was assessed by the Cervical Dystonia Impact Profile (CDIP-58) [16]. According to the United States Food and Drug Administration (FDA) guidelines for clinical trials, the CDIP-58, a patient-reported outcome measure (PROM), is commonly applied to evaluate cervical dystonia’s health impact since it has good reliability [19]. The CDIP-58 is composed of three conceptual domains, including (1) symptoms (3 subscales: head and neck, pain and discomfort, and sleep), (2) daily activities (2 subscales: upper limb activities and walking), and (3) psychosocial sequelae (3 subscales: annoyance, mood, and psychosocial functioning) [29,30,31]. A total score by summation of 8 subscales ranges from 58 to 290 points. A higher score represents a worse outcome. Since the Thai version of CDIP-58, compared with the original CDIP-58, has good reliability, we planned [13] to apply it to evaluate the disease-specific HRQoL at pre-treatment and post-two-year treatment after eight injections of Abo-BoNT-A [13].

#### 5.2.2. Secondary Outcome Measures

##### General Health-Related Quality of Life

The Short Form 36 health survey questionnaire (SF-36) was used for the evaluation of HRQoL.

The SF-36 comprises eight domains, including physical functioning (PF), role limitations due to physical health (RP), role limitations due to emotional problems (RE), vitality (VT), mental health (MH), social functioning (SF), bodily pain (BP), and general health (GH). Each domain is weighted by the sum of the questions in its section and directly transformed into a 0–100 scale. The lower the score, the greater the disability. Hence, a score of zero is equivalent to maximum disability. The Thai version of SF-36, validated and tested for reliability in CD patients [13], was applied to evaluate HRQoL in this study.

##### Depressive Symptoms

Center for Epidemiological Studies-Depression Scale

The CES-D questionnaire, a brief self-report scale, was developed to recognize depressive symptoms and severity in the general population. This twenty-item questionnaire asks about various symptoms of depression (depressed mood; feelings of guilt, worthlessness, and helplessness; psychomotor retardation; loss of appetite; and sleep difficulties) as they have occurred in the past week. The majority of the items focus on the affective component of depression and are arranged into six subscales reflecting the major symptoms of depression [32,33]. The CES-D score ranges from 0 to 3 (0 = rarely or none of the time (less than 1 day), 1 = some or little of the time (1–2 days), 2 = moderately or much of the time (3–4 days), 3 = most or almost all of the time (5–7 days)). Therefore, the sum of its scores ranges from 0 to 60. Higher scores represent more significant depressive symptoms. The cutoff scores of 20 or more (sensitivity = 79% and specificity = 80%) can categorize individuals with a risk of clinical depression. The Thai version of CES-D, validated and tested for reliability in Thai people [34], was used to assess depressive symptoms in this study.

Patient Health Questionnaire-9

The PHQ-9 [35], a reliable and validated measure, was used to assess depressive disorder in this study. The PHQ-9, based on the Diagnostic and Statistical Manual of Mental Disorders-IV-Text Revision criteria (DSM-IV-TR), is used to screen for a diagnosis of a depressive disorder. The total of all nine responses to the PHQ-9 aims to predict both the presence and severity of the depressive disorder. The scoring of each item is 0 (not at all), 1 (several days), 2 (more than half of the days), and 3 (nearly every day). Therefore, its total score ranges from 0 to 27 points: 5–9 points = minimal symptoms; 10–14 points = minor depressive disorder, dysthymia, or mild major depressive disorder; 15–19 points = moderate major depressive disorder; and 20 points or more = severe major depressive disorder.

### 5.3. Statistical Analysis

The demographic data were reported in terms of mean (standard deviation; SD) and range. The TWSTRS, CDIP-58, SF-36, CES-D, and PHQ-9 were presented as mean scores (standard error; SE) at baseline and at post 12- and 24-week treatments. Sample size was calculated before enrollment by non-inferiority statistical analysis of mean TWSTRS. The mean changes from baseline of both treatment groups were compared by using the paired *t*-test and Mann–Whitney U test with a significance level of α = 0.05 and beta error at 0.8. Additionally, the non-inferiority statistical analysis was used to compare mean reductions from baseline of both low-dose Neu-BoNT-A (50 units) and Abo-BoNT-A (250 units). All data were analyzed using the Cytel^®^ Studio^®^ (license no 2060107) software package. Sealed envelope randomization using online software applications for randomizing patients was performed in this clinical trial. **CLINICAL TRIAL REGISTER:** **NCT03805152**.

## Figures and Tables

**Figure 1 toxins-13-00694-f001:**
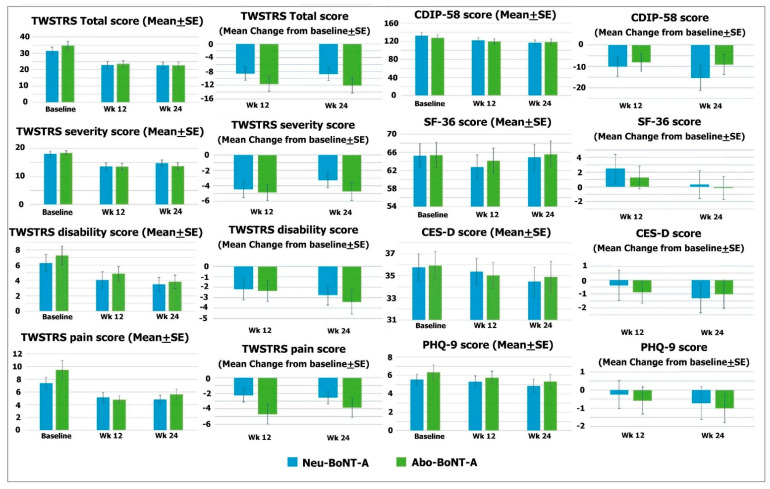
Graphs represent clinical score and mean change score before and after treatment as measured by TWSTRS, CIDP-58, SF-36, CES-D, and PHQ-9.

**Figure 2 toxins-13-00694-f002:**
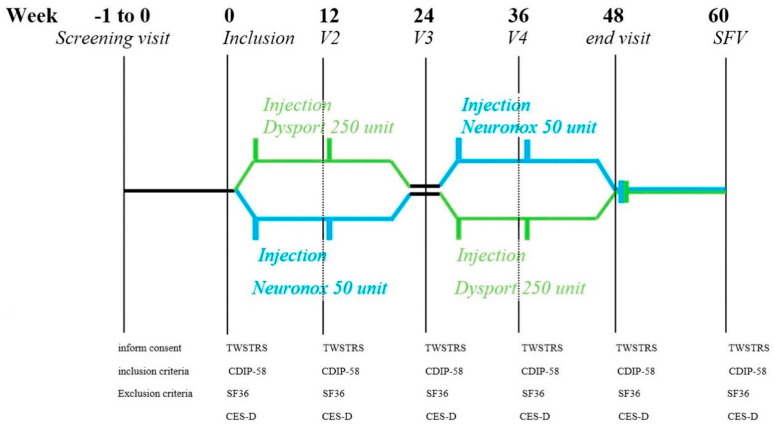
Study design: the 48-week prospective study of efficacy and quality of life in cervical dystonia patients before and after injections of low-dose Neu-BTX-A versus low-dose Abo-BTX-A at 12-week intervals over a 24-week treatment period, followed by crossover of low-dose Abo-BTX-A (250 units) versus low-dose Neu-BTX-A (50 units) at 12-week intervals over a 24-week treatment period. There was no washout period at week 24 in study design since duration effect of botulinum toxin A lasts for less than 12 weeks in cervical dystonia treatment. (TWSTRS—Toronto Western Spasmodic Torticollis Rating Scale, CDIP-58—Cervical Dystonia Impact Profile-58, SF-36—Short Form 36 health survey questionnaire, CES-D—Center for Epidemiological Studies-Depression Scale.)

**Table 1 toxins-13-00694-t001:** Clinical score and mean change score before and after treatment as measured by TWSTRS, CIDP-58, SF-36, CES-D, and PHQ-9.

Questionnaires	Treatment Groups	Score at Baseline Mean (SE)	Score at Week 12 Mean (SE)	Score at Week 24 Mean (SE)	*p*-Value after Treatment
Primary outcomes					
TWSTRS Total	Neu-BoNT-A	31.4 (2.4)	22.8 (2.2)	22.6 (2.0)	**0.008**
	Abo-BoNT-A	34.6 (2.7)	23.5 (2.0)	22.5 (2.1)	**0.002**
*p*-value between treatment groups		0.230	0.480	0.585	
TWSTRS severity	Neu-BoNT-A	18.0 (0.9)	13.6 (1.2)	14.8 (1.1)	**0.020**
	Abo-BoNT-A	18.4 (0.8)	13.5 (1.2)	13.7 (1.2)	**0.009**
*p*-value between treatment groups		0.417	0.834	0.155	
TWSTRS disability	Neu-BoNT-A	6.3 (1.1)	4.1 (1.0)	3.5 (0.8)	**0.014**
	Abo-BoNT-A	7.3 (1.2)	4.9 (0.9)	3.8 (0.8)	0.146
*p*-value between treatment groups		0.397	0.886	0.930	
TWSTRS pain	Neu-BoNT-A	7.4 (0.9)	5.2 (0.8)	4.8 (0.6)	0.126
	Abo-BoNT-A	9.5 (1.4)	4.8 (0.6)	5.6 (0.8)	**0.015**
*p*-value between treatment groups		0.116	0.166	0.123	
CDIP-58	Neu-BoNT-A	132.2 (6.6)	122.1 (5.6)	116.8 (5.8)	0.232
	Abo-BoNT-A	127.8 (5.9)	119.6 (5.8)	118.6 (6.4)	0.416
*p*-value between treatment groups		0.558	0.558	0.344	
Secondary outcomes					
SF-36	Neu-BoNT-A	65.24 (2.6)	62.74 (2.6)	64.9 (2.7)	0.731
	Abo-BoNT-A	65.38 (2.7)	64.12 (2.7)	65.5 (2.9)	0.911
*p*-value between treatment groups		0.670	0.824	0.583	
CES-D	Neu-BoNT-A	35.77 (1.1)	35.38 (1.2)	34.5 (1.3)	0.573
	Abo-BoNT-A	35.92 (1.2)	35.04 (1.1)	34.9 (1.4)	0.726
*p*-value between treatment groups		0.823	0.634	0.578	
PHQ-9	Neu-BoNT-A	5.58 (0.5)	5.33 (0.6)	4.9 (0.7)	0.226
	Abo-BoNT-A	6.35 (0.8)	5.77 (0.7)	5.4 (0.7)	0.633
*p*-value between treatment groups		0.095	0.296	0.916	

**Table 2 toxins-13-00694-t002:** Mean change score before and after treatment as measured by TWSTRS, CIDP-58, SF-36, CES-D, and PHQ-9.

Questionnaires	Treatment Groups	Mean Change at Week 12	Mean Change at Week 24
Primary outcomes			
TWSTRS Total	Neu-BoNT-A	−8.6 (1.8)	−8.8 (1.9)
	Abo-BoNT-A	−11.6 (2.1)	−12.1 (2.2)
*p*-value between treatment groups		0.284	0.129
TWSTRS severity	Neu-BoNT-A	−4.5 (1.0)	−3.3 (0.9)
	Abo-BoNT-A	−4.9 (1.0)	−4.7 (1.1)
*p*-value between treatment groups		0.514	**0.034**
TWSTRS disability	Neu-BoNT-A	−2.2 (1.0)	−2.8 (0.9)
	Abo-BoNT-A	−2.4 (0.9)	−3.4 (1.1)
*p*-value between treatment groups		0.758	0.375
TWSTRS pain	Neu-BoNT-A	−2.3 (0.8)	−2.6 (0.7)
	Abo-BoNT-A	−4.7 (1.2)	−3.9 (1.2)
*p*-value between treatment groups		0.244	0.314
CDIP-58	Neu-BoNT-A	−10.1 (4.5)	−15.4 (5.8)
	Abo-BoNT-A	−8.1 (4.2)	−9.2 (4.6)
*p*-value between treatment groups		0.740	0.240
Secondary outcomes			
SF-36	Neu-BoNT-A	2.5 (1.8)	0.3 (1.8)
	Abo-BoNT-A	1.3 (1.5)	−0.2 (1.5)
*p*-value between treatment groups		0.460	0.440
CES-D	Neu-BoNT-A	−0.4 (1.0)	−1.29 (1.0)
	Abo-BoNT-A	−0.9 (0.7)	−1.02 (1.0)
*p*-value between treatment groups		0.072	0.922
PHQ-9	Neu-BoNT-A	−0.3 (0.7)	−0.71 (0.9)
	Abo-BoNT-A	−0.6 (0.7)	−1.00 (0.7)
*p*-value between treatment groups		0.760	0.320

## Data Availability

Not applicable.

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
