# Peer review of "Low-Dose Neubotulinum Toxin A versus Low-Dose Abobotulinum Toxin A Injection for the Treatment of Cervical Dystonia: A Multicenter, 48-Week, Prospective, Double-Blinded, Randomized Crossover Design Study"

_toxins, 2021, doi:10.3390/toxins13100694_

Round 1

Reviewer 1 Report

Overall a very well-described study. The conclusion that low doses of either Neu- or Abo-BoNT A are similar in their ability to increase disease outcomes appears to coincide with the evidence presented. 

There are a few major adjustments needed.  1. The author contributions, funding, and other ending sections were not filled out. 2. The previous data given in the introductory paragraph starting on line 44, as well as the data in the paragraph starting on line 160, needs to include more specifics. Was a combination of BoNTs given? If so which ones? At what dose? 3. Explanation of the difference between the different botulism toxins mentioned would be beneficial to the reader's understanding.  4. Definition of Low, Normal, and High doses of each BoNT-A mentioned. This can help with further comparisons. 5. More explanation as to the choice of Abo-BoNT-A instead of Ona-BoNT-A, as you compared Neu-BoNT-A to Ona-BoNT-A rather than Abo-BoNT-A.  6. Information about previous exposure times/amounts of patients included in the study in the results or methods category - was previous exposure considered a significant factor? if not, why? If so, is 6 non-previously exposed patients enough to compare statistically?   Minor adjustments to spacing and formatting (especially around references, Figures 1 and 2 are named incorrectly, the list starting line 196, and the erroneous paragraph break between lines 289/290) as well as numbers (should be either all written out or all in numerical form). There are also minor grammatical errors that could change the meaning of certain sentences (e.g. line 39 - "could" is too uncertain of a term in this case if you are referencing information that shows BoNT does inhibit muscle contraction; lines 57-60 - the meaning is unclear; line 126 - it is unclear what both BoNT-As are different from; lines 133 & 135 - an incorrect plurality of "was/were").    Preferential Changes: Table 1 and Table 2 may serve better in the supplementary material. This will allow space to break up the graphs in "figure 2" and make them larger for better readability. If this is not possible or not in your interests, then moving the graphs to be with the data tables would also bring greater clarity to the information. I also feel the result of no adverse events from Neu-BoNT-A could be better emphasized as it is a very promising result, even if on a small scale.

Author Response

Response to Reviewers

Reviewer 1

  1. The author contributions, funding, and other ending sections were not filled out.

Response : The author contributions, funding, and other ending sections were added

  1. The previous data given in the introductory paragraph starting on line 44, as well as the data in the paragraph starting on line 160, needs to include more specifics.

Response : All specific information from references in Introductory paragraph is proper, can not be changed

       Was a combination of BoNTs given? If so which ones? At what dose?

Response :  No combination of BoNTs, it is crossover design indicate in manuscript

  1. Explanation of the difference between the different botulism toxins mentioned would be beneficial to the reader's understanding.

Response :  the difference between the different botulism toxins mentioned in Introductory paragraph there were same structure but manufactured by different company

  1. Definition of Low, Normal, and High doses of each BoNT-A mentioned. This can help with further comparisons.

Response : The data specify normal US FDA dosage of Ona BoNT-A approval was added, to specify the normal dosage recommended by US-FDA and this study use Ona-BoNT-A 50 unit

Added : dosage in Introduction part (high light with yellow) as followings:

According to a double-blind, randomized controlled trial, both BoNT-A (25th to 75th percentile range of 198 -300 Units) and BoNT-B

  1. More explanation as to the choice of Abo-BoNT-A instead of Ona-BoNT-A, as you compared Neu-BoNT-A to Ona-BoNT-A rather than Abo-BoNT-A.

Response : Compare of Neu-BoNT-A to Ona-BoNT-A had been study in reference 8 as indicate in Introduction so were do compare Abo-BoNT-A to Ona-BoNT in conversion ratio 1:5 in this study

  1. Information about previous exposure times/amounts of patients included in the study in the

results or methods category –

Response : the result had been showed as following Result paragraph

The mean (SD) duration of cervical dystonia was 3.1 (3.1) years, ranged 0-14 years, with the mean (SD) duration of previous cervical dystonia treatment was 2.8 (2.7) years. A total of 45 patients were previously treated with Abo-BoNT-A, one patient was previously treated with Neu-BoNT-A, and six patients were not ever treated with BoNT-A.

 Was previous exposure considered a significant factor? if not, why?

Response : No because the BoNT effect neuromuscular for 3 months and not change disease progression since it is just supportive treatment. So the previous treatment had no effect to disease all patient need to be treated as long term treatment

If so, is 6 non-previously exposed patients enough to compare statistically?  

Response : No because the BoNT effect neuromuscular for 3 months and not change disease progression since it is just supportive treatment. So the previous treatment had no effect to disease all patient need to be treated as long term treatment all treat and not previous treated still need to treat for continuous every 3 month

Minor adjustments to spacing and formatting (especially around references, Figures 1 and 2 are named incorrectly, the list starting line 196, and the erroneous paragraph break between lines 289/290) as well as numbers (should be either all written out or all in numerical form).

 Response : already changed as possible in new manuscript uploaded

There are also minor grammatical errors that could change the meaning of certain sentences

 (e.g. line 39 - "could" is too uncertain of a term in this case if you are referencing information that shows BoNT does inhibit muscle contraction;

Response : confirm no change

lines 57-60 - the meaning is unclear; line 126 - it is unclear what both BoNT-A are different from;

lines 133 & 135 - an incorrect plurality of "was/were").  

Response : I cannot find the line 57-60 and 133-135 mentioned in new manuscript in the system

 Preferential Changes: Table 1 and Table 2 may serve better in the supplementary material.

Response : confirm no change

 This will allow space to break up the graphs in "figure 2" and make them larger for better readability. If this is not possible or not in your interests, then moving the graphs to be with the data tables would also bring greater clarity to the information. I also feel the result of no adverse events from Neu-BoNT-A could be better emphasized as it is a very promising result, even if on a small scale.

Response : confirm no change

Reviewer 2 Report

I have carefully read the clinical trial proposed by the authors. I would like to thank them for this very important work and the care taken with the methodology.

However, some points of their analysis are problematic and prevent the results from being analysed in a suitable manner.

Here are some major elements to reconsider:
- whether it is an efficacy or non-inferiority study is not clear. the number of subjects needed should be calculated before the start of the study taking this fact into account. this should also be presented in the article
- since it is a crossover study, is a washout period necessary? whatever the answer, this should be specified.
- what are the characteristics of the population at inclusion (socio-demographic, medical)?
- a simple analysis can be carried out to find out by matched method whether one treatment is more effective than another (and the Mann Whitney test is not a matched test, it is the Wilcoxon test, to be corrected). nevertheless, this is not the standard of analysis for a crossover study, and an anova-type model should have been carried out. it should be specified whether the order in which the treatment was carried out is important
- details of the randomization should be provided

Other points:
- Figure 1 should detail in the legend the abbreviations
- cervical dystonia should be defined in the introduction

In conclusion, major elements of the analysis methodology should be included. I would be happy to read the manuscript again if the opportunity arises and to go further in the analysis of the results when these blocking points have been taken into account.

Author Response

Here are some major elements to reconsider:

- whether it is an efficacy or non-inferiority study is not clear. the number of subjects needed should be calculated before the start of the study taking this fact into account. this should also be presented in the article

Response :  statistical sample sized were calculated before the trial, the following was added in statistical part

Sample size was calculated before enrollment by non-inferiority statistical analysis of mean TWSTRS.

- since it is a crossover study, is a washout period necessary?

Response : No because the BoNT effect neuromuscular for 3 months and not change disease progression since it is just supportive treatment. So the previous treatment had no effect to disease all patient need to be treated as long term treatment all treat and not previous treated still need to treat for continuous every 3 month no need wash out period

whatever the answer, this should be specified.

Response : Under figure 1 was add the detail as followings

There was no washout period at week 24 in study design since botulinum toxin A duration effect lasting for less than 12 weeks in Cervical Dystonia treatment

- what are the characteristics of the population at inclusion (socio-demographic, medical)?

Response : No other significant medical conditions except cervical dystonia

All patients had no other significant medical conditions. Was added in Result demography

- a simple analysis can be carried out to find out by matched method whether one treatment is more effective than another (and the Mann Whitney test is not a matched test, it is the Wilcoxon test, to be corrected). nevertheless, this is not the standard of analysis for a crossover study, and an anova-type model should have been carried out.

Response : The At hoc analysis by  Mann Whitney test is suitable for multiple group comparison

 it should be specified whether the order in which the treatment was carried out is important

Response : there were no over carried effect in this design because the BoNT effect neuromuscular for 3 months and not change disease progression since it is just supportive treatment. So the previous treatment had no effect to disease all patient need to be treated as long term treatment all treat and not previous treated still need to treat for continuous every 3 month no need wash out period

- details of the randomization should be provided

Response : Sealed Envelope randomization by online software applications for randomizing patients were performed in this clinical trials. Added in Statistical part

Other points:

- Figure 1 should detail in the legend the abbreviations

Response : Added in figure 1

( TWSTRS-Toronto Western Spasmodic Torticollis Rating scale, CDIP-58- Cervical Dystonia Impact Profile-58, SF-36 Short Form 36 health survey questionnaire, CES-D- Center for Epidemiological Studies-Depression Scale)

- cervical dystonia should be defined in the introduction

Response Added in introduction

a common abnormal movement disorder, in which patients suffer from repetitive and/or sustained involuntary contractions of neck muscles, resulting in abnormal neck twisting and/or posture.
